Machine learning models and dimensionality reduction for improving the Android malware detection

Morán Pablo 1
http://orcid.org/0000-0002-5181-0199 Robles-Gómez Antonio 1 arobles@scc.uned.es
http://orcid.org/0000-0002-0619-8615 Duque Andres 2
Tobarra Llanos 1
http://orcid.org/0000-0002-4089-9538 Pastor-Vargas Rafael 1
1 Departamento de Sistemas de Comunicación y Control, Universidad Nacional de Educación a Distancia , Madrid , Spain
2 Departamento de Lenguajes y Sistemas Informáticos, Universidad Nacional de Educación a Distancia , Madrid , Spain
Alarcon-Aquino Vicente
Electronic publication date: 2024 Dec 23
Publication date: 2024
Volume: 10
Electronic Location ID: e2616
Received 2024 Sep 13; Accepted 2024 Nov 26
Copyright: © 2024 Morán et al.
Copyright year: 2024
Copyright holder: Morán et al.
License: This is an open access article distributed under the terms of the Creative Commons Attribution License, which permits unrestricted use, distribution, reproduction and adaptation in any medium and for any purpose provided that it is properly attributed. For attribution, the original author(s), title, publication source (PeerJ Computer Science) and either DOI or URL of the article must be cited.
License URL: https://creativecommons.org/licenses/by/4.0/

Keywords: Machine Learning algorithms, Random Forest, Supervised feature selection techniques, Feature filtering techniques, Predictive goodness metrics

Funding: CiberCSI UNED 2023–2024 LearnIoTOnCloud 2023-PUNED-0018 CiberGID innovation group This contribution is supported within the framework of the CiberCSI UNED research group with the research project 2023-2024 LearnIoTOnCloud (2023-PUNED-0018), as well as the CiberGID innovation group. The funders had no role in study design, data collection and analysis, decision to publish, or preparation of the manuscript.

==============================
Today, a great number of attack opportunities for cybercriminals arise in Android, since it is one of the most used operating systems for many mobile applications. Hence, it is very important to anticipate these situations. To minimize this problem, the analysis of malware search applications is based on machine learning algorithms. Our work uses as a starting point the features proposed by the DREBIN project, which today constitutes a key reference in the literature, being the largest public Android malware dataset with labeled families. The authors only employ the support vector machine to determine whether a sample is malware or not. This work first proposes a new efficient dimensionality reduction of features, as well as the application of several supervised machine learning algorithms for prediction purposes. Predictive models based on Random Forest are found to achieve the most promising results. They can detect an average of 91.72% malware samples, with a very low false positive rate of 0.13%, and using only 5,000 features. This is just over 9% of the total number of features of DREBIN. It achieves an accuracy of 99.52%, a total precision of 96.91%, as well as a macro average F1-score of 96.99%.

Introduction

With the evolution of technology in recent years, people have become hyper-connected (Al-Fuqaha et al., 2015). New technologies have opened up a wide range of very attractive possibilities, such as monitoring our day-to-day, providing intelligence to our home and controlling it from a mobile phone, accessing the Internet from countless mobile devices, working remotely from home, being able to make on-line purchases, and so on. Today, few people consider spending a day without their mobile phone or smart device.

All this modernization also carries a great risk, as that any device connected to the Internet is a potential target for cybercriminals (ISACA, 2019), questioning their security and, even worse, exposing our privacy to the world. We can suffer from the disabling of the device to theft of credentials or personal data, which can even lead to bank deposit robbery or identity theft over the Internet, among other threats.

Numerous operating systems (OS) have been developed for the management of smart devices, with Android and iOS being the two dominant brands in the market (Statcounter, 2024). This market was led by Microsoft years ago; however, it has been losing presence in favor of open-source systems. This trend has been marked by the above-mentioned boom in the use of mobile technologies that have overshadowed traditional PCs.

Reviewing the statistics on OS usage in the last years (Statcounter, 2024), the increase in popularity of Android can be visually observed to the detriment of Windows. This website is based on the existing data traffic through the Internet of mobile devices, tablets, PCs, and game devices. Android had a usage rate of 7.62% in July 2013, reaching 45.16% in September 2024, establishing itself as the leading operating system. It is followed by Windows, which, having a usage rate of 70.93% in July 2013, has been clearly affected by the rise of Android, and now represents 26.67% of the usage rate. We can also observe an slightly upward trend for iOS, although it is not as significant as for Android. This is because this OS is only used by Apple devices, which have a smaller number of terminals and gadgets.

The dominance of Android as the main mobile and tablet OS since 2013 can also be clearly observed (Statcounter, 2024). The last two years were marked by Android’s total control, with iOS in a more discrete second place. The rest of OS practically disappear from the graph. The upload and download peaks of Android match the minimum values for iOS. At the beginning of the last decade, Series 40 and SymbianOS seemed to keep the pulse with these two giants, but over the they collapsed over the years.

For all these reasons, Android OS is becoming the most desirable operating system for cybercriminals. Its intensive use opens up a wide range of possibilities to carry out non-ethical techniques against people who are not aware of the consequences of installing a malicious application (Shishkova & Kivva, 2022), such as theft of bank credentials, phishing, cyberextortion, etc. To minimize this issue, Google has strongly updated its security policy (Google, 2024). It has incorporated the use of machine learning (ML) based algorithms to detect possible malicious applications. However, despite all this, the risk that some “malicious” application ends up in the repository is always present.

Another growing problem is the existence of unofficial markets that offer countless open-source applications, many of them unrevised, which can expose our devices to possible infection. Android also offers a wide variety of options to manufacturers and developers, especially when it comes to patches and updates, resulting in a lack of control. For example, the Kotlin programming language (Kotlin, 2024) is quite popular for its concise code, faster development time, and its lighter learning curve (Putranto et al., 2020). This fact would allow hackers to develop dangerous mobile apps. However, as the use of mobile applications increased due to the COVID-19 pandemic, the number of existing threats in 2020 for mobile device applications increased considerably compared to the previous year (Shishkova & Kivva, 2022).

After careful examination of the situation described above and the different options on the market for analyzing Android applications, the opportunity to investigate the development of early malware detection methods, in which there is no need to install an application to detect if it is malicious or not, clearly arises. The increase in the use of machine learning-based techniques for this type of analysis has been one of the key triggers for the development of this study. The systematic review of several studies related to this field has given us the opportunity to establish a starting point for analyzing some of the less common aspects explored in previous works. A similar recent study based on static and online analysis using deep learning approaches can be found in Brown, Gupta & Abdelsalam (2024).

One of the most popular works focused on mobile devices is DREBIN (Arp et al., 2014; Spreitzenbarth et al., 2013; Machine Learning and Security, 2024), which presents a self-created dataset composed of both benign and malicious instances of mobile apps. DREBIN is a key reference in the literature. Due to its novelty in proposing a large domain-specific feature set that was manually engineered for the problem of Android malware detection (Daoudi et al., 2022). We chose to use the DREBIN dataset due to its complete labeling, which makes it perfectly suitable for training supervised classification models. Furthermore, DREBIN is widely known and used for its high accuracy and run-time efficiency (Garcia, Hammad & Malek, 2018; Narayanan et al., 2017). DREBIN’s authors also propose a first ML model for Android malware detection.

The main purpose of this work is to analyze in depth different methods of analysis for Android apps and to develop with this base a system capable of detecting whether an Android application is benign (goodware) or contains malware. Prior to any subsequent analysis, the data is subjected to a rigorous preprocessing procedure. This is followed by a hyper-parameter tuning of the ML models, with the latter carried out using a stratified 10-fold cross-validation approach. Our research goes beyond the quantitative measurements of several metrics, such as precision and recall, to analyze other qualitative dimensions of the approach in terms of families and key characteristics. In this context, a key contribution of this work is the analysis of the impact of applying dimensionality reduction techniques to the set of features used to perform this classification. Through these techniques, we aim to reduce the computational complexity of algorithms performing malware detection, which should lead to the development of systems more suitable for their use in contexts for which computational cost can be a major handicap (e.g., the use of these algorithms integrated with apps available on Android and IOs).

The main contributions of this research article are the following: Studying the evolution of the Android operating system, its advantages and disadvantages, its impact on society, and its possible security problems. In addition, several ML techniques for detecting malware in Android applications have been analyzed in search of malware.

Expanding the focus of the DREBIN project on early malware detection, using various machine learning algorithms, and assessing their behavior.

Studying the possible sets of discriminant features in ML algorithms and compare the results of using each of them.

Applying feature selection techniques to assess the impact of dimensionality reduction on the final results, through different evaluation metrics.

The remainder of the article is organized as follows. The “Background” section describes the state of the art of Android, its rise, its main advantages and disadvantages, and the impact of mobile virology on this OS. Several studies related to the static analysis of Android applications by applying machine learning techniques are also summarized. After that, the “Material and Methods” section details the experimental design, the dataset distribution and preprocessing used in the research, exposes different techniques for dimensionality reduction, and describes the machine learning algorithms used in the experimentation phase. Then, the “Performance Evaluation” section describes the experimental configuration, the metrics employed to measure the efficiency and effectiveness of each model, and the main results obtained for the different executions. Finally, our contributions and conclusions of this work are presented in the “Conclusions” section, as well as some future lines of work.

Background

Attack vectors

Malware can hide behind applications that users download to their smart devices. Sometimes, these applications can potentially damage these devices. Android is one of the most widely used operating systems for which the number of apps with behavior different from the expected has increased over time. It is essential to anticipate these situations; however, this is not an easy task because sample malware is not known at first. According to Shishkova & Kivva (2022), a total of 3,464,756 malicious installation packages, 97,661 new mobile Trojans banking and 17,372 new mobile ransomware were detected in 2020.

The main attack vectors of malware on our Android (Belcic, 2024) mobile devices are the following: Infected apps. Hackers take advantage of popular mobile applications to make a repackaged copy, in which the code is altered for malicious purposes. These applications are distributed through different unofficial markets. In other cases, these threats take the form of new applications that look entirely benign but are explicitly designed for non-ethical purposes.

Malicious advertising (Malvertising). Malware can be hidden behind ads and downloaded to the device after clicking on them. This is one of the most common methods of infection on Android devices.

Scams. This category includes phishing and other methods based on social engineering, through several communication options, such as e-mail. The message sent to the victim usually includes a malicious link that leads to the download of a specific application infected with malware.

Direct downloads from devices. This type of infection is more complex, as it requires another device containing the malicious application to be connected to the victim’s device. Other possibilities in this category usually imply physical access to the victim’s device. This strategy is often used for industrial intrusions.

Malware protection

Google is making substantial efforts to combat malware infections in Android. The permission system included in Android version 6 allows users to control accesses to the application to add restrictions. In addition to this, the publication of apps in Google’s official repository must meet a series of specific requirements. This is oriented towards the compatible development of applications with the latest versions of the operating system with more advanced layers of security.

Another option is the use of machine learning-based techniques to detect possible infectious applications. In this sense, The AppDefense Alliance (Ranchal, 2019) is a program created by Google in partnership with other security companies for Android malware detection. Google uses the shared data within its machine learning systems. On the other hand, Google (2024) is an advanced protection program devoted to increase the security of user accounts that require it. It enforces the security of users’ personal data and scans all applications installed on the Android device. Furthermore, SafetyNet (Xatakandroid, 2024) checks if the integrity of the analyzed device has been compromised after the installation of an app.

To promote the use of machine learning algorithms, the first step is to extract metadata and features associated with the app. In the case of Android applications, these data can be extracted, for example, from the general configuration file, called “manifest.xml”, which is contained inside the “.apk” executable file. This file contains relevant information, such as the execution permissions required to run the application, among other details. Direct analysis of this types of files are characterized by their low speed and usually do not have much success with obfuscated code. Therefore, it is essential to be able to disassemble the application code.

Table 1 shows a comparison of different approaches for Android malware detection, including our current proposal. Some indicators are presented, such as the employed dataset, the features used to feed the ML models, the inclusion or not of a process a dimensionality reduction, the specific ML algorithms selected for the malware detection, and the evaluation metrics. One of the most popular datasets on this topic is the free-to-download DREBIN (Arp et al., 2014) project. DREBIN is still one of the most accurate and best performing datasets. It uses a lightweight static analysis to extract eight different types of features from the bytecode of the application and the manifest file (Cao et al., 2020).

Table 1 Comparison of different DREBIN-based approaches for Android malware detection.

	Dataset	Features	Dimensionality reduction	Machine learning algorithms (used in each of the works)	Metrics	
DREBIN (Arp et al., 2014)	Self-created (123,453 benign apps; 5,560 malware)	All (“manifest.xml”/Disassembled code of apps). The eight subset of features with the corresponding permissions are detailed in Table 2	No	Support vector machine	Recall, False positive rate	
MLDroid (Mahindru & Sangal, 2021a)	Based on DREBIN and self-created features	Features only related to permissions and API calls	Yes: Feature ranking and feature subset selection	Naive Bayes, Logistic regression, Decision trees, Nonlinear decision tree forest (Random Forest), Support vector machines, and others	Accuracy, F1-score, Precision, Recall	
HybriDroid (Mahindru & Sangal, 2021b)	Based on DREBIN and self-created features	Features only related to permissions and API calls	Yes: T-test analysis + Multivariate linear regression stepwise forward selection	Logistic regression, Artificial Neural network, Radial basis function neural network, Best training ensemble, Majority voting ensemble, Random forest	Accuracy, F1-score, Precision, Recall	
EC2 (Chakraborty, Pierazzi & Subrahmanian, 2020)	Based on DREBIN and Koodous	All features	Yes: Manually selected features	Decision tree, K-nearest neighbors, logistic regression, Naive Bayes, Support vector machine, Random forest, DBSCAN, Hierarchical, Affinity, K-Means, MeanShift	Precision, Recall, F1-score, Area under the curve	
AOMDroid (Jiang et al., 2020)	Based on DREBIN and the benign apps (verified by VirusTotal)	All features (with obfuscation)	Yes: TF-IDF based algorithm	Transfer learning models	Accuracy	
Factorization-Machine-based proposal (Li et al., 2019)	Based on DREBIN and AMD	Features from manifest files and source code	Yes: Manually selected features	Support vector machines, Naive Bayes, Multi-layer perceptrons, Factorization machine	Accuracy, Precision, Recall, F1-score, False positive rate, Training time	
DroidFusion (Yerima & Sezer, 2019)	Malgenome-215, Drebin-215 (Yerima, 2018), McAfee-350, McAfee-100	Drebin-215: Only 215 features divided into four primary categories: API calls; Permissions used in the “manifest.xml” file of DREBIN; Intents located at the “manifest.xml” file of DREBIN; Command patterns The work also employs Malgenome-215, McAfee-350 and McAfee-100	Yes: Manually selected features	Random forest, REPTree, Random Tree-100, Random Tree-9, Voted perception, Majority voting, Average of probabilities, Maximum probability, MultiScheme, DroidFusion	True positive rate, False positive rate, Precision, F1-score, Time	
SecuDroid (Vanusha et al., 2024)	Drebin-215 (Yerima, 2018; Yerima & Sezer, 2019)	Based on Drebin-215, by including some transformations	No: Drebin-215 is already reduced	Decision tree, Support vector machines, Logistic regression, K-nearest neighbors, SecuDroid	Accuracy, Precision, Recall, F1-score, Area under the curve	
Authors’ proposal	Based on DREBIN	All (“manifest.xml”/Disassembled code of apps). The eight subset of features with the corresponding permissions are detailed in Table 2	Yes: Chi-Squared	Naive Bayes, Logistic regression, Decision trees, Random forest, Support vector machines	Accuracy, Precision, Recall, F1-score, False positive rate, Time	

Table 2 Dataset distribution for the DREBIN project (data type, feature type, permission type).

	Data type	Feature location	Possible permissions	
Subset 1	Hardware components	General “manifest.xml” file	Hardware accesses	
Subset 2	Request permissions	General “manifest.xml” file	Data and resources permissions	
Subset 3	Application components	General “manifest.xml” file	Activities, Services, Content providers, Broadcast receivers	
Subset 4	Filtered intents	General “manifest.xml” file	Communication intents	
Subset 5	Restricted API calls	Disassembled code	API calls	
Subset 6	Used permissions	Disassembled code	Read permissions	
Subset 7	Suspicious API calls	Disassembled code	API calls	
Subset 8	Networks addresses	Disassembled code	IP addresses, Host-names, URLs	

The authors of DREBIN performed an extensive analysis, using the features extracted from different Android applications. These features are included in a joint vector space in which a machine learning algorithm can be used to recognize a series of patterns and automatically determine whether an application is potentially malware or not. They used a self-created dataset that contained 123,453 benign applications (goodware) and 5,560 malware samples. They employed all types of features, both from the general “manifest.xml” files and those disassembled from the code of the applications. The eight subsets of features with their corresponding permissions are detailed in Table 2. Some results obtained in the study conducted by the DREBIN project were better than those obtained by other approaches on similar datasets, detecting 94% of the malware (recall) with a minimum false positive rate of around 1%. No dimension reduction process was performed for feature selection, thus, computational complexity can be a problem.

In Daoudi et al. (2022), an exploratory analysis of DREBIN is performed with the aim of revealing insights about how and why it works. The authors corroborate that the performance of the DREBIN dataset is stable and sustainable, becoming a strong reference that researchers should consider when assessing the performance of malware detection approaches. Their importance in the final predictions suggests that more research is needed on feature engineering that captures maliciousness, as well as further research on the design of quality algorithms and metrics for classifiers, beyond classical studies.

As detailed in Table 1 and presented in next sections, this research work takes the DREBIN dataset and project as a starting point, considering both types of features. However, in our case, an exhaustive dimensionality reduction process is carried out in order to reduce this complexity. In addition to this, several ML algorithms are employed, and an exhaustive performance evaluation is applied, obtaining promising results for the employed metrics.

Another study related to malware analysis in Android applications using ML techniques is MLDroid (Mahindru & Sangal, 2021a). It proposes a web-based framework to help detect malware on Android devices. MLDroid only extracts the features related to permissions and API calls. The results obtained are mainly based on the average F1-score values. In the case of our work, additional metrics are proposed with better results. Our average F1-score considers both malware and goodware classes in the classification outcome. On the other hand, dimensionality reduction in MLDroid is proposed as a step prior to the application of machine learning algorithms over the dataset. In our case, the feature selection is only performed on the training dataset, this way creating a more realistic experimentation framework, in which no test samples are used for this process, avoiding the potential bias that these samples may introduce in the final results.

HybriDroid (Mahindru & Sangal, 2021b) presents an effective model of malware analysis using ensemble methods. The proposal and conclusions are very similar to the previous one in terms of dataset development and dimensionality reduction. Other recent studies have also contributed extensively to the development of numerous contributions, due to the high availability and high-performance of DREBIN, as detailed in the EC2 (Chakraborty, Pierazzi & Subrahmanian, 2020) and AOMDroid (Jiang et al., 2020) proposals to predict Android malware families. A malware detector based on a Factorization Machine is also analyzed in Li et al. (2019).

On the other hand, a learning algorithm to enhance the security of linear models with DREBIN (Demontis et al., 2019) shows that performance can be deteriorated for attacker manipulation. In Li et al. (2022), the backdoor attack against Android malware detectors is studied. Another interesting work is DroidSieve (Suarez-Tangil et al., 2017), which considers only static features that are resistant to advanced obfuscation strategies. To minimize this impact, DroidCat (Cai et al., 2019) extracts dynamic features based on method calls and the invocation of sensitive APIs.

A recent approach is SecureDroid (Vanusha et al., 2024). This study examines five machine learning classification algorithms through the lens of a meticulous hyper-parameter tuning methodology and a comprehensive stratified four-fold cross-validation approach. The results are based on the evaluation of several key performance indicators, including accuracy, precision, recall, F1-score, and the area under the curve. All empirical validations were based on the Drebin-215 dataset (Yerima, 2018; Yerima & Sezer, 2019). The dataset comprises 215 features, which can be classified into four primary categories: 1) API calls, which are identifiers for methods derived from the APK’s.dex file; 2) Permissions delineated in the “manifest.xml” file of DREBIN, which outline the privileges requested by the APK; 3) Intents situated in the “manifest.xml” file of DREBIN. The “xml” file of DREBIN denotes the APK’s intentions for inter-component or inter-application communication; and, additionally, 4) Command patterns reference Linux commands found within APK files, including .dex, .jar, .so, and .exe.

This work begins with the results obtained by DREBIN, in which the complete dataset was used to train the model. The objective is to test the impact of dimensionality reduction through feature selection methods for this dataset. A more refined methodology from SecureDroid (Vanusha et al., 2024) was employed for our experimental design, which included precise hyperparameter tuning and evaluation through stratified 10-fold cross-validation. Furthermore, our approach also aims to expand the methodology employed by DREBIN from an algorithmic perspective. We propose the utilization of multiple machine learning algorithms to facilitate a comparative analysis of their respective outcomes.

Materials and Methods

Experimental design

This section addresses the characteristics of the dataset used in this study, providing a detailed overview of the distribution of malware and goodware samples, the breakdown of sample counts by family, the classification of features into different levels according to their type and the adjustments made to prepare the dataset for the experiments. These factors are crucial to understanding how the data were structured and processed, ensuring that the conducted experiments are valid and reproducible.

Following this, the theoretical foundation of the methodology used in the study is discussed in depth. Specifically, various dimensionality reduction techniques are analyzed, focusing on filtering methods that best align with the specific requirements of this research. The choice of these techniques is informed by the dataset’s characteristics, which include a mixture of categorical and numerical data and the need to develop models that can effectively handle this data structure. Filtering methods, such as Chi-square and mutual information, were prioritized for their ability to efficiently reduce the number of features while maintaining a balance between computational cost and model performance. Unlike more complex techniques such as wrapping or embedded methods, which involve running models iteratively to evaluate feature subsets, filtering methods are computationally lighter and, therefore, more scalable for large datasets like the one used in this work.

Furthermore, the study employs several supervised machine learning classification algorithms, which are carefully selected and compared based on their suitability for malware detection tasks. These algorithms, including decision trees, support vector machines, and random forests, were chosen for their proven effectiveness in handling high-dimensional datasets and their ability to model complex relationships between features. Each algorithm offers different advantages depending on the data’s specific nature and the study’s objectives. For instance, decision trees provide a clear interpretable advantage, whereas random forests offer robustness to overfitting, and support vector machines are highly effective in cases of nonlinear separability. By comparing the performance of these algorithms, this study aims to identify the most suitable classifier for the given problem, thus optimizing detection accuracy and computational efficiency.

A methodology for the experimental design with precise hyper-parameter tuning and evaluation through stratified 10-fold cross-validation has been employed for all experiments. With regard to dimensionality reduction, several subsets of features have been selected (50; 500; 5,000; 15,000) for the five machine learning algorithms employed here, following a testing and tuning of the preprocessing phase. For each type of experiment, results are analyzed considering first only the features of the “manifest.xml” file for dimensionality reduction and, secondly, all features included. The evaluation of the tested models is based on their accuracy, precision, recall, F1-score, and false positive rate on an unseen test dataset. The specific metrics to be used will be defined later. Our findings provide a comparative analysis of various machine learning models for malware detection and introduce a computationally efficient approach.

Dataset distribution

In this work, the dataset provided by the DREBIN project is taken as a starting point to decide whether a sample is malware or not. The main reasons for selecting DREBIN are varied: it is the largest public and labeled mobile malware dataset in the literature (Chakraborty, Pierazzi & Subrahmanian, 2020), and it contains specific features manually engineered for Android malware detection. Moreover, the dataset presents a fairly generic feature set, considered relevant for the detection of different malware samples over time (Daoudi et al., 2022). This is especially important when considering the sustainability of models trained with this dataset. According to the initial information, extracted from the DREBIN documentation, the dataset contains 123,453 benign applications and 5,560 malware samples, i.e., a total of 129,013 samples. After compiling all the features extracted from the files and eliminating duplicates, a total of 545,333 features are obtained. The dataset is distributed in several folders containing the samples of the applications, the characteristics of each application in a “.txt” file and several additional folders with the test sets used in the DREBIN experiments.

Recent research works on the DREBIN dataset has consistently demonstrated its long-term significance as a benchmark within the scientific community for malware analysis on Android platforms. A collection of machine learning methods, including SVM, Random Forest, logistic regression, decision trees, K-nearest neighbors, and ensemble methods are tested in Jyothsna et al. (2024), Vanusha et al. (2024), Surendran et al. (2024), together with some feature selection techniques. Deep learning techniques such as deep convolutional networks have also been tested (Dong, Shu & Nie, 2024), and more sophisticated techniques involving ensemble learning have been applied in these networks (Xu et al., 2024). The continued use of DREBIN underscores its importance in evaluating and developing new approaches to malware detection. This is particularly relevant given the proliferation of Android devices and the increasing sophistication of attacks targeting this operating system. Android, as highlighted in earlier sections, has emerged as the dominant mobile platform, widely surpassing alternatives such as Windows and iOS. This prominence is due not only to its large user base, but also to the unique landscape of cybersecurity challenges, making it a focal point for malware research. Given these factors, DREBIN remains an indispensable tool for researchers aiming to assess and improve malware detection techniques in the context of Android.

The primary objective of the present study is to extend the work done on the DREBIN dataset by introducing a novel approach to malware detection. This approach seeks to improve upon existing methods by implementing advanced techniques for data processing and dimensionality reduction. These techniques aim to improve the precision and efficiency of malware classification without compromising computational performance. Moreover, the proposed methodology is designed to be highly adaptable, allowing its application beyond the Android ecosystem to other operating systems or security environments. This adaptability ensures that the findings of this study contribute not only to the enhancement of Android malware detection, but also to the broader field of cybersecurity research.

Figure 1 shows the distribution of features in percentage, which is divided by type of feature (section “manifest” indicates features from the “manifest.xml” file, while section “dis_code” shows features extracted from the disassembled code), as well as by the target class: goodware (GW) or malware (MW). The distribution of the DREBIN dataset with respect to the types of existing data, feature type, and type of permission required to run the application is given in Table 2. We can observe how all data related to hardware components, request permissions, application elements and filtered intents belong to the general features. Furthermore, data related to restricted and suspicious calls, permissions used, and network addresses are features of the internal code itself.

Figure 1 Feature distribution by type and class (percentages).

Figure 1 also shows that the dataset employed is heavily unbalanced, due to the difficulty of finding malware samples. The proportion of malware and goodware samples is maintained throughout all the experiments developed in this work, for all the training and testing splits considered. Specific metrics are also used in the evaluation of these experiments to ensure an appropriate treatment of these imbalance issues, as will be explained in subsequent sections.

Data preprocessing

The DREBIN dataset does not consist of a simple “.csv” file. Instead, it is distributed in several folders containing the samples of the applications, the characteristics of each app in a “.txt” file and several more folders with the test sets they used in the DREBIN experiments. A list labeling the samples as malware or goodware is not available, so this information had to be extracted from the test sets manually. All features extracted from the samples, both from the “manifest.xml” file and from the decompiled application code itself, were used as input data.

According to the initial information, extracted from the DREBIN documentation, the dataset has 123,453 benign and 5,560 malware applications, i.e., a total of 129,013 samples. Combining all features extracted from the files and removing duplicates gives a total of 545,333 features.

In order to be able to label the samples, a script is developed for relating the applications with the classes to which they belong (malware/goodware), extracting the information from the test set. Once the data had been labeled, we found that there was no information on 67 samples, so we decided to remove them from the dataset. Finally, the dataset used in this work is made up of 123,393 goodware samples and 5,553 malware samples (128,946 samples in total). After these adjustments, the total number of features is recalculated to 545,196.

After the above steps, a binary matrix is created for which rows represent applications and columns represent features. This matrix is filled by default to 0, and a 1 is entered if the application X contains the feature in column Y. The hash of the application is used as the application identifier.

The code used and the procedure for reproducing these experiments can be found in our GitHub repository, as well as the results obtained summarized in a “.csv” file.

Dimensionality reduction

Dimensionality reduction, when it comes to machine learning, can be defined as the transformation process of input variables from a high-dimensional to a low-dimensional space (Van Der Maaten, Postma & Van den Herik, 2009). In general, this reduction of the number of dimensions within an input instance usually helps to fight the so-called “Curse of Dimensionality” (Bellman, 1961), which states that the number of samples needed to successfully estimate a function grows exponentially with respect to the number of dimensions in the input variables. This leads to important problems such as the risk of overfitting or a very high computational cost when training machine learning models. In order to avoid these problems, a dimensionality reduction process has been performed. In this way, the most relevant characteristics are used for classifying a sample. At the same time, it will help us to analyze the impact of dimensionality reduction on the results obtained, compared to the results obtained when using all possible features (Brownlee, 2020). As shown in Table 1, many different studies can be found in the literature regarding the use of dimensionality reduction techniques in the context of cybersecurity and malware detection in particular. Some of these techniques lead systems to better results in terms of detection, accuracy, and false positive rates (Mohammadi et al., 2019). Regarding the DREBIN dataset, some recent studies have shown the existence of redundant and even useless features within the dataset, indicating that a thorough dimensionality reduction process can lead systems to achieve similar results to those obtained by using the whole set of features in their training process (Daoudi et al., 2022).

There are many dimensionality reduction techniques: feature selection methods try to automatically select the most relevant features by scoring methods or statistical values and discard the rest. The main classes of this type are wrapper methods and filtering methods. Other techniques such as matrix factorization are based on linear algebra and aim to reduce the matrix that makes up the dataset, subsequently using this subset as input data in the model. The most common approach according to matrix factorization is principal components analysis (PCA). Multiple learning techniques are usually unsupervised methods that create low-dimensional projections from high-dimensional data while preserving the most outstanding relationships between the data. The most commonly used multiple learning methods are Kohonen self-organizing maps (Kohonen, 2001) or t-SNE (Van der Maaten & Hinton, 2008). Finally, autoencoder techniques are neural models based on an encoder and a decoder that produce nonlinear projections of the original input into a low-dimensional space.

In the case of this work, since the target variable is found and labeled in the dataset used for performing the experiments, we will focus our attention on supervised feature selection techniques. Hence, the first goal will be to find out whether each variable within the sample instances is informative enough to predict the target variable (malware or not). Supervised feature selection techniques are classified as follows: Filter methods. These methods focus on analyzing whether there are statistically significant relationships between the input (independent) and output (dependent) characteristics. Once the results are obtained, they are ranked according to their confidence level, from higher to lower. This ranking process is independent of the specific model that will ultimately be used, which means that the results of these evaluations are not influenced by any particular learning or prediction model. One of the main advantages of these methods is their ability to introduce controlled biases to minimize the risk of overfitting, which helps improve the model’s generalization to new data. Furthermore, these methods are computationally more efficient compared to wrapping methods, as they require fewer resources and less processing time, making them suitable for applications where computational cost is a significant concern.

Wrapping methods. Wrapping methods are based on training particular models with different subsets of features, with the final purpose of selecting those subsets that offer the best results in an inference step. These methods are able to detect interactions between features, so the results obtained are usually more accurate. However, they present a higher computational cost and are prone to overfitting. Wrapping methods can be based on forward selection or backward elimination, that is, iteratively adding the feature that best improves the model or removing the least significant feature for the model, respectively, until convergence is reached. Greedy optimization algorithms such as recursive feature elimination (RFE) (Guyon et al., 2002) are also employed as wrapping methods.

Intrinsec/embedded methods. Although embedded methods are similar to wrapping methods, they take into account a target function of a prediction model, hence the feature selection process is integrated within the final learning algorithm. Although they do not usually have high computational cost or overfitting issues, they are highly dependent on the final model to be used. Some commonly used embedded methods are Lasso (L1) (Tibshirani, 1996) and Ridge (L2) (Hoerl & Kennard, 1970) regularization, which tend to shrink many of the features to values close to zero.

Input variables are those that provide data input to the model. They define the particularities of an instance. On the other hand, output variables are those that “label” an instance. They are considered dependent variables of the input (independent) variables. The values of these variables are the values that the model is aiming to predict.

This work has been based on filtering techniques, mainly because of their lower computational cost and their robustness to overfitting. Moreover, filtering techniques do not depend on the model used. Basically, each input variable is classified according to its score in relation to the target variable and then selected or eliminated from the data set. The choice of the statistical model used to perform this scoring is directly dependent on the type of data of the variables. The most common types of variables are numerical (integer or floating) and categorical (nominal, ordinal and Boolean or dichotomous). It is therefore important to be familiar with the types of data of the variables in the dataset or whether it is possible to convert them to another type that facilitates the calculations, keeping in mind that the nature of the data is not altered. The type of output variable dictates the type of predictive modeling problem that needs to be built. For the most common data types, numerical and categorical, these would be regression-predictive modeling problems for the former and classification for the latter.

The most common predictive models are the following: Pearson’s correlation coefficient. It is the most popular correlation coefficient between two input and output numerical variables. Pearson’s coefficient measures the ratio between the covariance of two variables and the product of their individual standard deviations. It is commonly used in linear regression to measure how strong the relationship is between the two variables. The result returns a value between −1 and 1, where 1 indicates full positive correlation, and 0 indicates no correlation. Negative values indicate a negative growth in one variable when the other grows positively, and vice versa.

Spearman’s rank correlation coefficient. It is a non-parametric measure in which the monotonicity of the relationship between two variables is evaluated. In this coefficient, the variables are ranked before performing a calculation similar to the Pearson’s coefficient. Normality is not assumed for the considered distributions. The meaning of the correlation coefficient is similar to Pearson’s.

Analysis of variance (ANOVA). It is a parametric test of statistical hypothesis, used to determine if two or more data samples belong to the same distribution, by calculating the ratio between variance values. An F-test or F-statistic is employed for the statistical analysis, and those input variables found to be independent from the output variable are discarded. The ANOVA test is used when the input variable is numeric and the output variable is categorical, or vice versa.

Kendall T coefficient. It is is a non-parametric test used to measure the ordinal association between two ranks. Considering two variables X and Y and the pairs of observations (xi,yi) and (xj,yj) with i<j, a pair is said to be concordant if either xi<xj and yi<yj or xi>xj and yi>yj, this is, if the sort order of xi and xj agrees with that of yi and yj. The coefficient measures the difference between the number of concordant and discordant pairs, and is normalized using the total possible number of pairs. The correlation coefficients vary between −1 (no association) and 1 (identical variables).

Chi-squared. The Chi-squared test determines the dependency between two variables, and it is only applicable to categorical and nominal variables. The coefficient uses observed values for measuring deviation from the expected distribution, considering that the predictor variable and the target variable are independent. A null hypothesis test is established, which will be accepted if the obtained scored is below a threshold and rejected otherwise. Hence, if the test is accepted, the two variables will be considered to be independent, and hence the predictor variable will be excluded from the dataset, since it cannot be used for predicting the output. If the test is rejected, the predictor variable will be maintained, as its dependency towards the target variable may be important for the classification model. For the calculation of the coefficient, only non-negative features and classes are allowed. The test result produces scores for each feature, allowing a ranking and determining the most significant features for the evaluated dataset. Therefore, it fits with the data and models proposed in this work, as a feature selection method.

Mutual information. Mutual information is a symmetric value that indicates the dependency between two random variables, obtaining a non-negative value, by measuring the reduction of uncertainty about Y having observed X and vice versa. The mutual information score approaches zero in case both variables are independent. It supports target variables that are continuous or discrete. The function is based on non-parametric methods that rely on entropy estimation from distances calculated with the k-nearest neighbors method.

As mentioned above, the Chi-square test was selected to reduce the final dimensionality of the original dataset. This choice is primarily motivated by the nature of the data in this research, where both the features and the target variable are binary and, therefore, categorical. The Chi-square test is explicitly designed to evaluate relationships between categorical variables, making it an ideal fit for this dataset. Its strength lies in determining whether a statistically significant association exists between the observed distributions of the variables and their expected distributions under the assumption of independence.

In contrast, other statistical tests like Pearson’s or Spearman’s correlation coefficients are more suited for continuous or ordinal numerical data, and are therefore inappropriate for this type of binary-categorical analysis. Pearson’s correlation, for example, measures the strength of a linear relationship between two continuous variables, while Spearman’s correlation is used to assess monotonic relationships, both of which assume that the data have a natural order or continuous scale. Similarly, ANOVA tests are typically used when comparing the means of numerical attributes across different categories, making them a better fit for datasets where the features are continuous but the target variable is categorical. Likewise, the Kendall-Tau coefficient, often applied to continuous or ordinal variables, is not appropriate for binary-categorical datasets.

Given these considerations, the Chi-square test was chosen not only because of its relevance to categorical data but also because of its efficiency in handling a large number of features during the reduction of dimensionality. It allows for a simple and computationally inexpensive assessment of the statistical significance of feature-target relationships without making assumptions about the distribution of the data. This makes it particularly useful in scenarios where the goal is to filter out features that do not contribute meaningful information to the target variable, ultimately helping to create a more efficient and effective model.

Classification algorithms

This section will describe the algorithms used for this work (Pedregosa et al., 2011). Specifically, we have focused on supervised algorithms (Buitinck et al., 2013), in order to compare the results obtained with the DREBIN project and other related works described in the background section. The selection of the optimal hyperparameters for each model was based on the accuracy, precision, recall, F1-score, and false positive rate obtained from the 10-fold cross-validation process. The incorporation of the F1-score was intended to offset potential biases that might otherwise arise from class imbalance, while the remaining metrics served as a basis for evaluating the efficacy of each classifier (Vanusha et al., 2024). Once the optimal hyperparameters had been identified, the performance of each classifier was subjected to a rigorous test on the unseen test dataset.

These algorithms have also been selected according to the binary nature of the input and target variables: Naïve Bayes (NB) (Zhang, 2004). It is a classification algorithm based on Bayes’ theorem. This theorem describes the probability of occurrence of an event based on previous similar conditions. The algorithm is based on the assumption that the features are not related among them. It has been widely used in popular tasks such as spam filtering, sentiment analysis, and article ranking, among others.

It is one of the fastest and simplest machine learning algorithms. It can be used for binary predictions or multi-class target variables, and it tends to perform good multi-class predictions. Its main disadvantage is that it assumes that all variables are independent; hence, potential interrelationships between them are not considered.

A multinomial NB classifier has been applied in this work. The key hyperparameters in tuning it are alpha and the prior probability estimation method. Smoothing priors, such as those with alpha greater than 0, account for features not present in the training data and prevent zero probabilities in subsequent calculations. In our case, we employed Laplace smoothing, where alpha is set to 1. The class prior probabilities will be estimated from the training data.

Logistic regression (LR) (Nocedal, 1980). It is a classification algorithm for dichotomous variables in which one or multiple independent variables can be used to predict an outcome. In addition to predicting the target class, this algorithm also indicates the probability of belonging to a given class. With respect to binary classification, the model demonstrates enhanced performance compared to linear regression. Linear regression establishes as class-0 all the samples with probability less than 0.5, where those with probability greater than or equal to 0.5 are class-1. Logistic regression improves this problem by applying a sigmoid function to smooth the line that separates the samples from each class.

Its main advantage is the ease of extension to multiple class predictions. Furthermore, it is a rapid classification algorithm with unidentified records and is less susceptible to overfitting. As main disadvantages, it can suffer from overfitting if the number of samples is less than the number of features. Linearity between the dependent and independent variables is also assumed. Only discrete functions can be predicted through this technique.

The key hyperparameter in tuning a logistic regression algorithm is the regularization strength, known as C. A higher C reduces regularization, potentially leading to overfitting. Conversely, a lower C increases regularization, potentially improving generalization. We explored various C values using stratified 10-fold cross-validation and found that C = 1 was optimal for our model. On the other hand, parameter “penalty” specifies the type of regularization technique employed for reducing the risk of overfittin. In our case, we employ L2 regularization. Finally, the “lbfgs” (Limited-Memory BFGS or Broyden–Fletcher–Goldfarb–Shanno) algorithm is employed as the solver used in the optimization process.

Decision trees (DTs) (Quinlan, 1990). In this technique, the pathway leading to a classification is represented by a tree, in which the internal nodes represent features, and decisions about these features are represented by branches. These can lead to another internal node or a leaf, which shows the result of the prediction. It simulates a graphical representation of decision making, obtaining all possible solutions under a set of conditions. This algorithm usually imitates the human-thinking ability to make a decision, easily understood thanks to its tree-like structure.

Its main advantages are its ease of understanding, its usefulness for solving decision-making problems and interpreting all the possible results of a problem, and also the reduced need for data cleansing when compared to other algorithms. However, overfitting is very usual. It can also be complex with many layers and class labels.

The key hyperparameters in this case are the maximum depth of the tree, the minimum number of samples required in a leaf node, the splitting criterion, and the minimum number of samples required to split an internal node. In our experiments, we have allowed nodes to expand until all leaves are pure or all leaves contain fewer than two samples. This means that each leaf node will contain at least one sample, and that the minimum number of samples required to split an internal node is 2. The Gini index was selected as the criterion to measure the quality of a split.

Random Forest (RF) (Biau, 2012). Random Forest is a supervised algorithm that can be used in both classification and regression problems. It is based on the concept of ensemble learning, which is a process of combining several classifiers to solve a complex problem to improve the performance of the model. This classifier is composed of several decision trees that are fed with random subsets of the training dataset. To select the final prediction result, the algorithm is based on the voting of the predictions. A high number of trees improves accuracy and prevents overfitting. It is widely used and successfully in different domains such as banking, medicine, or marketing. Among its main advantages, it is capable of handling high-dimensional datasets, it can be used in both classification and regression tasks, and it is capable of increasing precision and avoiding overfitting. Its main disadvantage is that it is not suitable for regression.

The key hyperparameters in tuning the Random Forest algorithm are the number of estimators (number of trees in the forest), the splitting criterion, the maximum depth of the tree, the minimum number of samples required in a leaf node, and the minimum number of samples required to split an internal node. In our experiments, we have allowed nodes to expand until all leaves are pure or all leaves contain fewer than two samples. This means that each leaf node will contain at least one sample, and that the minimum number of samples required to split an internal node is 2. The number of estimators is 100, and the Gini index was selected as the criterion to measure the quality of a split.

Support vector machine (SVM) (Cortes & Vapnik, 1995; Scholkopf & Smola, 2018). It is an algorithm that can be used both for classification and regression; however, it is mainly used for classification problems. Data are represented in an n-dimensional space, and the main goal of the algorithm is to find a hyperplane capable of optimally separating the points into different spaces based on their class. Data points from the different classes that are closer to the hyperplane, and influence its position and orientation, are called support vectors.

There are two types of SVM; linear, for those cases in which linear separability for each data point of the dataset is assumed, and hence the algorithm tries to maximize the distance from the data points to the hyperplane; and nonlinear otherwise. Non-linear SVMs rely on nonlinear kernels for transforming the feature space so the hyperplane can then be found. SVMs are often used for image analysis, handwritten digit recognition, text mining, spam detection, and outlier detection and clustering.

As main advantages, SVMs present great precision in high n-dimensional spaces and are efficient in memory management. However, they are prone to overfitting if the number of features is greater than the number of samples, no direct probability estimates are provided, and they are not very computationally efficient for large datasets.

The key hyperparameters for this algorithm are the regularization parameter (C), the kernel function, the degree of the polynomial kernel (if used), and the gamma parameter of the RBF kernel (if used). We employed our SVM algorithm with C=1, an RBF (Radial Basis Function) kernel. Since the kernel is not polynomial, no degree is specified, and the gamma value is assigned to ‘scale’ which automatically determines the optimal gamma value based on the data.

Table 3 summarizes the values assigned to the most important hyperparameters for each of the explored machine learning algorithms.

Table 3 Hyperparameter values for the different machine learning methods explored.

Algorithm	Hyperparameters	
Naïve bayes	α=1.0	
	Prior probabilities = None	
Logistic regression	C=1.0	
	Penalty = L2	
	solver = “lbfgs”	
Decision tree	Max. Depth = Non set (maximum possible)	
	Min. Samples leaf = 1	
	Min. Samples split = 2	
	Quality criterion = “Gini”	
Random forest	Max. Depth = Non set (maximum possible)	
	Min. Samples leaf = 1	
	Min. Samples split = 2	
	No. Estimators = 100	
	Quality criterion = “Gini”	
SVM	C=1.0	
	Kernel function = RBF	
	gamma = scale	

Performance evaluation

This section details the configuration setup designed to carry out the set of experiments, the definition of metrics employed in this study to calculate the performance of the different tests and the obtained results, together with the improvements obtained in relation to previous works.

Experiment setup

To carry out the experimentation phase for this work, a laptop with an Intel Core i7 8750H processor, 250 GB M2 NVE, 1 GB SSD and 32 GB of RAM has been used. The software used has been Jupyter Labs, with a programming environment in Python 3 and open-source Python libraries for the implementation of the proposed models (Pedregosa et al., 2011).

The work has been divided into two main experiments, which have been in turn separated into two subexperiments. The first one, named “Experiment 1” (E1), consists of a training and testing model using cross-validation of 10 iterations. These iterations are performed on the same dataset, configuring 90% of the samples to train the model and 10% to perform the test. It should be noted that the samples used for testing are never repeated among iterations. Figure 2 shows the flow chart of this experiment configuration, including several subsets of characteristics (50; 500; 5,000; 15,000) for the five machine learning algorithms employed here.

Figure 2 Flow diagram for Experiment 1 (E1), consisting of both training and test phases for each ML algorithm and each selected amount of features.

A total of 10 iterations for cross-validation purposes. E1-A only considers features of the “manifest.xml” file for dimensionality reduction; E1-B considers all features this selection.

The second experiment, named “Experiment 2” (E2) represents a more exhaustive environment. It consists of a cross-validation process of 10 iterations with a training and test splits, and a final validation dataset. In this case, 20% of the dataset samples are reserved for validation tests and the remaining 80% is used to perform the 10-fold cross-validation. Once the 10 iterations for each algorithm and set of features have been performed, the best model is selected for the subsequent validation process. Figure 3 shows the different tasks in the flow chart of this experiment. The ratio of goodware-malware samples within every split employed in this experiment was assured to be similar. The main objective is to emulate an experimental environment as realistic as possible, in which the selected training model would be used to deploy the malware detection system.

Figure 3 Flow diagram for Experiment 2 (E2), consisting of both training and test phases for each ML algorithm and each selected amount of features, as well as a final validation phase.

A total of 10 iterations for cross-validation purposes. It considers both features from the “manifest.xml” file and disassembled features for dimensionality reduction.

Each of both experimental configurations (E1 and E2), is performed twice, one of them with the general features contained in the “manifest.xml” and the another one with all features. Therefore, the second sub-experiments take into account the dynamicity of the disassembled source code. Samples are tagged and sorted by their hash code, so there is no possibility of altering the partitions based on the name given to the sample. Moreover, as previously mentioned, the proportion of malware and goodware samples considered in each training and testing subsets involved in the experiments is maintained so a proper extrapolation of the obtained results can be done.

To ensure a robust evaluation, each classifier was trained using its optimal hyperparameters, determined through a meticulous tuning process conducted solely on the training dataset. The test dataset, kept entirely separate, was not exposed to the models during the training phase. This rigorous approach was adopted to mirror real-world scenarios, where models encounter unseen data. By adhering to this methodology, we aimed to provide a reliable and accurate assessment of the models’ potential performance in practical applications.

Metrics

Different metrics have been used in this work to evaluate the effectiveness of the developed models. Before this, some definitions are needed, as detailed next: True positive (TP). In this work, the metric refers to the samples that are correctly classified as malware by the prediction model.

True negative (TN). In this work, the metric refers to the samples that are correctly classified as goodware by the prediction model.

False positives (FP). In this work, the metric refers to the goodware samples that are incorrectly classified as malware by the prediction model.

False negative (FN). In this work, the metric indicates the number of malware samples that are incorrectly classified as goodware by the prediction model.

The specific metrics employed in this work are accuracy, precision, recall, Avg F1-score and false positive rate. They are detailed in the following paragraphs.

Accuracy. This metric measures the relationship between the samples that were correctly classified (TN and TP) and the total amount of samples. Accuracy is calculated as detailed in Eq. (1).

(1) Accuracy=(TN+TP)(TN+FN+TP+FP)

Precision (P). This metric measures the ratio between the samples classified with a specific class (goodware or malware) and those that actually belong to the class itself. Precision is calculated as detailed in Eqs. (2) and (3) for the goodware and malware classes, respectively.

(2) Precision(GWclass)=TN(TN+FN)

(3) Precision(MWclass)=TP(TP+FP)

Recall (R). This metric measures the ratio of correctly classified cases for each class. In the case of this work, we also deal with the recall for both goodware and malware classes. Recall is calculated as detailed in Eqs. (4) and (5) for the goodware and malware classes, respectively.

(4) Recall(GWclass)=TN(TN+FP)

(5) Recall(MWclass)=TP(TP+FN)

Avg F1-score. This metric is an average score of the F1-score for each considered class (malware and goodware). The F1-score is the harmonic mean of precision and recall. Hence, the average F1-score takes both classes into account, and is detailed in Eq. (7).

(6) F1−score=2∗P∗RP+R

(7) AvgF1−score=(F1GWclass+F1MWclass)2

False positive rate (FPR). This metric is calculated by dividing the total number of samples incorrectly classified as malware (False positives or FP), between the total number of goodware samples (TN + FP). It is calculated as detailed in Eq. (8).

(8) FPR=FP(TN+FP)

Results

In this section, all results obtained in the proposed experiments are shown and discussed. Although all the previously mentioned metrics have been calculated, only the most relevant ones are shown to compare the different algorithms considered. Precision and Recall of class “Malware” illustrates the proportion of malware samples correctly identified, since the main purpose is to find a model that minimizes errors related to detecting malware samples. Additionally, the accuracy metric shows the correctness of a model, as detailed above. It is also interesting to consider the rate of false positives (goodware samples classified as malware), which should be kept to a minimum. Additionally, macro Avg F1-score is used to illustrate the overall performance of the system. Figure 4 shows the evolution of the macro Avg F1-score as the total number of features used to train the model increases.

Figure 4 Macro Avg F1-score as a function of the total number of features considered.

Results using only features from the “manifest.xml” file (left graph) or using all features (right graph), after a 10-fold cross validation process. Meanings of the acronyms denoting each different algorithm can be found in a previous section.

As mentioned previously, the Chi-squared test is carried out for reducing the final number of features, which is set to four different possible values: 50; 500; 5,000; and 15,000 features. Figure 4 shows how all models rapidly converge to a similar macro Avg F1-score (when compared to the same algorithm) when the number of features reaches the value of 5,000, while the differences between using 5,000 and 15,000 features are usually negligible. Naïve Bayes is the only algorithm that performs slightly different, since it obtains its best results when using 500 features. This can be due to the simplicity of the model, which leads to a worsening of the overall performance when a high number of features is considered, probably because of the noise introduced by those new features. However, this behavior of the Naïve Bayes method should be considered in the general context, in which the results achieved by this technique are always worse than those obtained by the rest of the algorithms considered. All these aspects can be seen both in the experiments using only features from the “manifest.xml” file and in the experiments using the whole feature set.

Regarding particular results, Table 4 shows the best results for each of the considered algorithms in experiment E1-A (average results for a 10-fold cross validation), when considering only the features included in the “manifest.xml” file. The results of the same experiment using the entire set of features as initial input (E1-B) are illustrated within Table 5. Scores are shown for the best feature selection configuration for each algorithm.

Table 4 Experiment 1-A (E1-A).

Features from the “manifest.xml” file. Best feature selection configuration for each algorithm, indicated in row “Features (best value)”. Bold indicates best results for each of the considered metrics.

	NB	LR	DT	RF	SVM	
Features (best value)	5,000	15,000	5,000	5,000	5,000	
Accuracy	97.93%	99.23%	99.34%	99.42%	99.40%	
Precision (MW class)	71.86%	95.71%	94.22%	96.80%	96.35%	
Recall (MW class)	85.40%	85.95%	90.37%	89.46%	89.49%	
Macro Avg F1-score (MW/GW classes)	88.46%	95.08%	95.95%	96.34%	96.23%	
FPR	1.50%	0.17%	0.25%	0.13%	0.15%	

Table 5 Experiment 1-B (E1-B).

All features considered. Best feature selection configuration for each algorithm, indicated in row “Features (best value)”. Bold indicates best results for each of the considered metrics.

	NB	LR	DT	RF	SVM	
Features (best value)	500	15,000	5,000	5,000	5,000	
Accuracy	96.86%	99.33%	99.42%	99.52%	99.45%	
Precision (MW class)	59.56%	95.34%	93.73%	96.91%	96.57%	
Recall (MW class)	83.69%	88.93%	92.79%	91.72%	90.43%	
Macro Avg F1-score (MW/GW classes)	83.95%	95.83%	96.47%	96.99%	96.95%	
FPR	2.55%	0.20%	0.28%	0.13%	0.14%	

The data in both tables demonstrates that an increase in the number of features used in training results in a notable enhancement in overall performance. In general, the best results are obtained using 5,000 features, both for the experiment considering only features from the “manifest.xml” file and for the experiment considering the whole set of features. The only algorithms that do not exactly follow this trend are Naïve Bayes with the entire set of features, in which using 500 features offers better performance, and LR for both experiments, which obtains its best results with 15,000 features.

The technique achieving the best results is Random Forest for all the metrics considered except for recall of the “Malware” class, in which the best performing algorithm is DT. Random Forest is able to overcome the results from the rest techniques for precision of the “Malware” class, macro average F1-score and false positive rate (in this metric, smaller values indicate better performance).

Considering the use of the entire set of features, which include features extracted from the disassembled code of the considered applications, all models increase their malware detection ability when using all the possible features, except for naïve Bayes, also probably due to the simplicity of the technique and the noise introduced in the model when considering a more complex set of features. For the rest of the algorithms, macro Avg F1-score and recall of the “Malware” class always increase from experiment E1-A to E1-B, while precision of the “Malware” class and FPR show some fluctuations, showing better performances for the most complex algorithms (RF and SVM). In general, the use of these more complex classification algorithms and more extensive feature sets leads to better results in this experiment, however, a trade-off between performance and resource consumption is always needed for determining whether the improvement of results compensates for the higher use of resources required by these algorithms.

Table 6 shows a comparison of the best results achieved by each algorithm tested in the experimental framework defined for experiment E2, this is, separating a subset of 20% of the dataset for testing purposes, performing a 10-fold cross-validation training and testing on the remaining 80%, and selecting the best model (fold) for each algorithm in this step for the final test.

Table 6 Experiment 2 (E2).

Comparative summary among obtained results with both the “manifest.xml” file and all features. Type of feature configuration indicated in row “Type of feature”. Best feature selection configuration for each algorithm, indicated in row “Features (best value)”. Bold indicates best results for each of the considered metrics.

	NB	LR	DT	RF	SVM	
Type of feature	“manifest.xml” file	All features	All features	All features	All features	
Features (best value)	5,000	15,000	15,000	5,000	5,000	
Accuracy	97.89%	99.38%	99.42%	99.52%	99.47%	
Precision (MW class)	70.58%	95.66%	94.36%	97.04%	96.55%	
Recall (MW class)	87.27%	89.53%	92.06%	91.70%	90.97%	
Macro Avg F1-score (MW/GW classes)	88.47%	96.08%	96.45%	97.02%	96.70%	
FPR	1.63%	0.18%	0.24%	0.13%	0.15%	

The scores shown in the table do not imply significant differences with respect to Tables 4 and 5, that is, results of E1-A and E1-B. In general, all algorithms except naïve Bayes perform better when using the entire set of features for training, and the number of final features for which convergence is reached is 5,000, except for the LR and DT techniques. Regarding metrics, Random Forest obtains the best overall results in terms of accuracy, precision (“Malware” class), macro Avg F1-score and FPR, while DT overcomes it in terms of recall of the “Malware” class (with 15,000 features).

The comparison with the most recent work using the DREBIN dataset to analyze the effectiveness of different models in malware detection is quite complex, mainly due to differences in the methodology followed. One of the most similar recent studies (Dong, Shu & Nie, 2024) applies deep learning techniques, specifically deep convolutional neural networks (D-CNN), for malware classification, also using the DREBIN dataset. The best results reported by this study show an accuracy of 96.72%, an F1-score measure of 93.9%, and a false positive rate (FPR) of 0.45%. As seen in Table 6, the results obtained by our best configuration (Random Forest with 5,000 features extracted from the total set of features) outperform all these metrics.

Although dimensionality reduction is used in some of these works, none of them carry out an analysis as exhaustive as the one presented in this research, where various subsets of features are analyzed, either extracted from the original set of features or solely from the features in the “manifest.xml” file. For example, in the previously mentioned study (Dong, Shu & Nie, 2024), the Ant Colony Optimization (ACO) (Dorigo & Stützle, 2004) method is mentioned as a feature selection technique; however, the final number of features used for classification is not specified. This also happens in other works: specifically, in Jyothsna et al. (2024) the Grey Wolf Optimization (GWO) algorithm (Mirjalili, Mirjalili & Lewis, 2014) is mentioned without further details on the number of extracted features; and in Xu et al. (2024), the Chi-Square test is applied to extract a fixed number of features without investigating the impact of different feature set sizes.

Finally, it is important to highlight the main differences between some previous works showing good results on malware detection and our approach. For instance, accuracy and F1 scores of around 99% are reported in Jyothsna et al. (2024), Xu et al. (2024) and Vanusha et al. (2024). In our case, according to the results obtained by our system, we are able to achieve similar results in terms of accuracy, although our best performing configuration remains two percentage points below that score in terms of F1-score. However, all those approaches make use of a version of the DREBIN dataset named DREBIN-215 (Yerima, 2018; Yerima & Sezer, 2019), which contains a total of 5,560 malware apps and 9,476 benign apps, for which a fixed set of 215 features is proposed. Therefore, the task addressed in these works is substantially different from the one presented in this article, and hence, comparisons are not entirely fair due to differences in the distribution of instances belonging to the “malware” and “benign” or “goodware” classes, as well as the total number of features considered. In our work, as discussed in the “Data preprocessing” subsection, we use a total of 5,553 malware samples and 123, 452 benign apps, along with a feature set composed of 545,196 features. It is noticeable that, even considering this particularity, our system is still able to overcome all those systems in terms of FPR. In particular, the best FPR value for our system is 0.13%, while a FPR of 0.92% is reported in both (Jyothsna et al., 2024; Xu et al., 2024).

Conclusions

The first objective of this research is to study the impact of Android applications on society and, additionally, their possible security issues. These apps can become attack vectors for malware. After reviewing the state-of-the-art in malware analysis with machine learning techniques, the model presented by the DREBIN project was found to show very promising results in terms of malware detection and false positive rate. An elaborated dataset of features was made from both the “manifest.xml” file and the disassembled code applications, while most of the existing approaches are solely based on the general features.

It should be noted, however, that the initial DREBIN approach was based solely on the SVM algorithm and considered all features. As part of our main contribution, this fact led us to extend their approach with additional models (such as naïve Bayes, logistic regression, decision trees and Random Forest) and assessment metrics, as well as introducing techniques for exploring dimensionality reduction of features to minimize overfitting and computation complexity of the experiments. The Random Forest model has proven to obtain the most promising results to date. In particular, it achieves an accuracy of 99.52%, a malware detection precision of 96.91%, a macro Avg F1-score of 96.99% (considering both malware and goodware classes), and a false positive rate of only 0.13%.

The best-performing models developed in this research work only use around 9% of the features after having selected the most relevant ones during the dimensionality reduction process. This significantly reduces the computational cost and time required, making our models more readily applicable to devices with limited resources. Our proposed model improves by far the false positive rate of 1% of the DREBIN project (7.7 times lower), as well as the rest of the recent proposals compared in the first part of this study. On the other hand, the malware detection rate (recall) of the proposed model is close to the score of 94% offered by the DREBIN project. An experiment emulating a real use case has also been proposed, leading to even better results when using Random Forest techniques with 5,000 selected features.

In addition, comparing our models with MLDroid research, the authors also found that the Random Forest algorithm is the best-performing model. Their F1-score for the malware class only reaches 94% twice for some specific applications. The rest of the experiments scored below 90% for this metric. In our work, the average F1-score (malware and goodware) values are much higher in most cases, especially in the experiments executed with the selected Random Forest model, which, additionally, presents a much stronger dimensionality reduction. The conclusions obtained compared to the HybriDroid approach are very similar.

Our contribution presents limitations that must be considered for further works. As a purely static analysis, it does not have the ability to analyze possible application transformations at runtime. However, it can anticipate some cases by analyzing obfuscation-related features. Due to the difficulty in finding recent malware samples, there is a lack of such samples in the dataset, especially regarding certain specific families. The provision of this information would facilitate enhancements to the robustness and accuracy of the model. Another limitation of static analysis is the possibility of mimetic and poisoning attacks. Renaming the activities and components of an application between the training and detection phases can affect discriminative features. On the other hand, it is also possible to bypass the analysis result by adding features that are typical of goodware applications.

In future work, it is planned to extend our study by adding additional ML algorithms, such as neural networks, and to compare their performance with the results obtained in this research study. Another line of research will include a second level of feature selection, using wrapper methods, to detect potential relationships between the different existing features. Additionally, the analysis of the most relevant features extracted for each of the subsets generated with the selected dimensionality reduction algorithm is one of the main future directions of this work. Despite its significant implications, given the size of the feature subsets, this more detailed qualitative study can provide beneficial and interesting insights about the most informative characteristics for detecting malware in Android applications.

We want to thank the authors of the DREBIN project (Arp et al., 2014; Spreitzenbarth et al., 2013; Machine Learning and Security, 2024), for sharing their dataset to be used as a basis in this work. The authors would also like to acknowledge the Spanish Government (through the Cybersecurity Institute of Spain, INCIBE), with the Strategic Research Project “Analysis of mobile applications from the perspective of data protection: Cyber-protection and Cyber-risks of citizen information” and the International Chair “Smart Rural IoT and Secured Environments”, within the context of the Recovery, Transformation and Resilience Plan financed by the European Union (NextGenerationEU/PRTR). Both the Strategic Research Project and the International Chair were granted in a public call by INCIBE in 2023. INCIBE is an entity dependent on the Spanish government. This is one of the first works related to both initiatives, and it will lead to further research in the field of AI, cybersecurity, and mobile devices.

Additional Information and Declarations

Competing Interests

Author Contributions

Data Availability

The authors declare that they have no competing interests.

Pablo Morán performed the experiments, analyzed the data, performed the computation work, prepared figures and/or tables, and approved the final draft.

Antonio Robles-Gómez conceived and designed the experiments, analyzed the data, performed the computation work, prepared figures and/or tables, and approved the final draft.

Andres Duque conceived and designed the experiments, analyzed the data, performed the computation work, prepared figures and/or tables, and approved the final draft.

Llanos Tobarra conceived and designed the experiments, analyzed the data, prepared figures and/or tables, authored or reviewed drafts of the article, and approved the final draft.

Rafael Pastor-Vargas conceived and designed the experiments, analyzed the data, prepared figures and/or tables, authored or reviewed drafts of the article, and approved the final draft.

The following information was supplied regarding data availability:

The code, data origin, and results are available at GitHub and Zenodo:

- https://github.com/PabloMT-UNED/Machine-learning-models-and-dimensionality-reduction-for-improving-Android-malware-detection.

- Morán, P., Robles-Gómez, A., Duque, A., Tobarra, L., & Pastor-Vargas, R. (2024). Machine learning models and dimensionality reduction for improving Android malware detection. Zenodo. https://doi.org/10.5281/zenodo.14034272.

The dataset is property of the DREBIN project, which gave permission to carry out the experiments, but we are not authorized to share the material. Please request access to the dataset at: https://drebin.mlsec.org.

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
