# Peer review of "Machine learning models and dimensionality reduction for improving the Android malware detection"

_PeerJ Computer Science, doi:10.7717/peerj-cs.2616_

## Round 0.1 · original submission · Major Revisions

I have received reviews of your manuscript from scholars who are experts on the cited topic. They find the topic interesting; however, several concerns regarding experimental setup and comparisons with previously reported approaches must be addressed. These issues require a major revision. Please refer to the reviewers’ comments at the end of this letter; you will see that they advise you to revise your manuscript. If you are prepared to undertake the work required, I would be pleased to reconsider my decision. Please submit a list of changes or a rebuttal against each concern when you submit your revised manuscript.

Thank you for considering PeerJ Computer Science for the publication of your research.

With kind regards,

·

Basic reporting

The manuscript is written in clear and professional English, making it easy to follow. The author provides a comprehensive background and context, referencing existing literature effectively. The article is well-structured, with figures and tables that support the narrative. The results are self-contained and relevant to the hypothesis, with clear definitions of terms used throughout the study. The clarity and detail make the work reproducible.

Experimental design

The research is original and relevant to the journal's scope, focusing on Android malware detection. The research question is well-defined and meaningful, addressing a gap by improving upon DREBIN with fewer features. The investigation is thorough, adhering to high technical and ethical standards. The methods are described in detail, allowing for replication. The author explains the selection of datasets, dimension reduction, and machine learning algorithms comprehensively, comparing existing methods and justifying their choices.

Validity of the findings

The findings are robust and statistically sound, with a thorough evaluation using different machine learning models. The author demonstrates how fewer features achieve slightly better performance than previous methods. While the novelty is clear, additional insights into why the selected features are effective would strengthen the paper, despite the challenge posed by the large number of features. The conclusions are well-stated and linked to the original research question, supported by the results.

Additional comments

The paper presents a convincing study on Android malware detection using machine learning. The clarity of the writing and thoroughness of the methodology are commendable. Providing insights into the selected features' effectiveness, while challenging, would enhance the paper. Overall, the research is impressive and contributes valuable findings to the field.

Reviewer 2 ·

Basic reporting

The format can be improved.

Experimental design

The method section must be more clear and it must be given in detail. There must be more comparisons with the other recently proposed methods in order to judge the impact and difference of the newly proposed approach.

Validity of the findings

There must be more comparisons with the other recently proposed methods in order to judge the impact and difference of the newly proposed approach. There must be more references from the latest research on this area/topic.

Additional comments

The paper should be carefully revised by a native English speaker or a professional language editing service to improve the grammar and readability.

---

## Round 0.2 · Minor Revisions

All concerns raised by the reviewers have been partially addressed. The manuscript still needs further clarification regarding comparisons with previously reported approaches and the hyperparameter selection for the ML models. A table with hyperparameters should be provided. These issues have not been addressed and therefore a minor revision is required. If you are prepared to undertake the work required, I would be pleased to reconsider my decision. Please submit a list of changes or a rebuttal against each point that is being raised when you submit your revised manuscript.

·

Basic reporting

no comment

Experimental design

no comment

Validity of the findings

no comment

Additional comments

The author has addressed all my concerns.

---

## Round 0.3 · accepted · Accept

I am pleased to inform you that your work has now been accepted for publication in PeerJ Computer Science.

Please be advised that you cannot add or remove authors or references post-acceptance, regardless of the reviewers' request(s).

Thank you for submitting your work to this journal. I look forward to your continued contributions on behalf of the Editors of PeerJ Computer Science.

With kind regards,